# *TNF*-Block Genotypes Influence Susceptibility to HIV-Associated Sensory Neuropathy in Indonesians and South Africans

**DOI:** 10.3390/ijms21020380

**Published:** 2020-01-07

**Authors:** Jessica Gaff, Fitri Octaviana, Prinisha Pillay, Huguette Gaelle Ngassa Mbenda, Ibnu A. Ariyanto, June Anne Gan, Catherine L. Cherry, Peter Kamerman, Simon M. Laws, Patricia Price

**Affiliations:** 1School of Pharmacy and Biomedical Science, Curtin University, Bentley 6102, Australia; jessica.gaff@postgrad.curtin.edu.au (J.G.); juneanne.gan@postgrad.curtin.edu.au (J.A.G.); Peter.Kamerman@wits.ac.za (P.K.); s.laws@ecu.edu.au (S.M.L.); 2Neurology Department, Faculty of Medicine, Universitas Indonesia, Jakarta 10430, Indonesia; fitri.octaviana@gmail.com; 3Neurology Department, Cipto Mangunkusumo Hospital, Jakarta 10430, Indonesia; 4Brain Function Research Group, School of Physiology, University of Witwatersrand, Johannesburg 2193, South Africa; prinishapillay_13@yahoo.com (P.P.); ngassagaelle@yahoo.fr (H.G.N.M.); kate.cherry@monash.edu (C.L.C.); 5Virology and Cancer Pathobiology Research Center, Universitas Indonesia, Jakarta 10430, Indonesia; ibnu.ariyanto07@gmail.com; 6Department of Infectious Diseases, Alfred Health and Monash University, Melbourne 3004, Australia; 7Burnet Institute, Melbourne 3004, Australia; 8Collaborative Genomics Group, School of Medical and Health Sciences, Edith Cowan University, Joondalup 6027, Australia

**Keywords:** HIV, sensory neuropathy, *TNF*-block, 8.1 ancestral haplotype, *DDX39B* and *BAT1*

## Abstract

HIV-associated sensory neuropathy (HIV-SN) is a disabling complication of HIV disease and antiretroviral therapies (ART). Since stavudine was removed from recommended treatment schedules, the prevalence of HIV-SN has declined and associated risk factors have changed. With stavudine, rs1799964*C (TNF-1031) associated with HIV-SN in Caucasians and Indonesians but not in South Africans. Here, we investigate associations between HIV-SN and rs1799964*C and 12 other polymorphisms spanning *TNF* and seven neighboring genes (the *TNF*-block) in Indonesians (*n* = 202; 34/168 cases) and South Africans (*n* = 75; 29/75 cases) treated without stavudine. Haplotypes were derived using fastPHASE and haplotype networks built with PopART. There were no associations with rs1799964*C in either population. However, rs9281523*C in intron 10 of *BAT1* (alternatively *DDX39B*) independently associated with HIV-SN in Indonesians after correcting for lower CD4 T-cell counts and >500 copies of HIV RNA/mL (model *p* = 0.0011, Pseudo *R*^2^ = 0.09). rs4947324*T (between *NFKBIL1* and *LTA*) independently associated with reduced risk of HIV-SN and shared haplotype 1 (containing no minor alleles) associated with increased risk of HIV-SN after correcting for greater body weight, a history of tuberculosis and nadir CD4 T-cell counts (model: *p* = 0.0003, Pseudo *R*^2^ = 0.22). These results confirm *TNF*-block genotypes influence susceptibility of HIV-SN. However, critical genotypes differ between ethnicities and with stavudine use.

## 1. Introduction

HIV-associated sensory neuropathy (HIV-SN) is a disabling complication of HIV disease and its treatment. It predominately affects nerve fibers that innervate the distal limbs, particularly the feet [1]. Symptoms include pain, burning and numbness, which impact an individual’s quality of life and ability to work [2,3]. Nucleoside reverse transcriptase inhibitors (NRTI) are effective anti-retroviral therapies (ART) but some, most notably stavudine (d4T), have severe adverse effects, including sensory neuropathy and lipodystrophy [4]. The prevalence of HIV-SN varied from 19% to 57% in patients exposed to stavudine [5,6,7,8,9]. Stavudine has not been recommended since 2010 due to its toxicity and a reduction in HIV-SN cases have been noted but not well documented.

We compared the prevalence of HIV-SN assessed with the AIDS Clinical Trial Group Brief Peripheral Neuropathy Screening Tool in HIV+ patients treated at an inner-city clinic in Jakarta, Indonesia with stavudine in 2006 and without stavudine in 2016 [10]. A patient’s height and age were associated with HIV-SN among patients receiving stavudine, and 34% experienced neuropathy [5]. In 2016, the prevalence of HIV-SN was 14.2% among patients who had not been exposed to stavudine. Most patients (94%) had <500 copies HIV RNA/mL plasma, but >500 copies HIV RNA/mL and a nadir of below 200 CD4 T-cells/µL associated with HIV-SN. Thus, HIV-SN still presents in patients treated without stavudine, but the prevalence is lower, and the risk factors are markers of HIV disease severity. With the same screening test, the prevalence of HIV-SN was higher (57%) in South African patients receiving stavudine, and was linked with age and height, as in Indonesia [9]. The prevalence of HIV-SN without stavudine in this setting is unclear.

Tumor necrosis factor alpha (TNF) expression has been demonstrated histologically in specimens from individuals with sensory fiber, axonal and/or demyelinating neuropathies, particularly when there was pain [11]. Its role in HIV patients is supported by evidence that the application of HIV gp120 to the sciatic nerve of a rat upregulated expression of TNF, CXCR4 and CXCL12 in the dorsal root ganglia and the lumbar spinal dorsal horn. A non-replicating herpes simplex virus vector encoding the *p55TNFSR* gene (producing a TNF-soluble receptor able to block TNF bioactivity) reversed mechanical allodynia [12]. Since HIV patients display elevated levels of TNF and its soluble receptor in plasma [13], a role in HIV-SN is plausible. However, TNF is difficult to visualize in skin biopsies—perhaps because expression is transient. Genetic associations provide a signature that is stable over time.

The gene encoding TNF (*TNF*) lies in the central MHC in a defined region of high linkage disequilibrium (LD; the *TNF*-block), which contains several potentially pro-inflammatory genes (*TNF*, *LTB*, *LTA*, *NCR3*, *LST1*, *NFKBIL1*, *ATP6V1G2*, *BAT1* (alternatively *DDX39B*) and *MCCD1*) [14]. Carriage of the minor allele at TNF-1031 (rs1799964) associated with increased risk of HIV-SN in Caucasians, Chinese and Malays who had received stavudine [5,6,15], but not in South Africans [16]. We characterized *TNF*-block haplotypes in multiple ethnicities and showed that the haplotype containing the minor allele of TNF-1031 in Asians and Caucasians was not present in Africans [14,16]. This implicates an allele carried in linkage with TNF-1031 in its associations with HIV-SN.

Here, Indonesian and South African HIV patients treated without stavudine are assessed to identify single nucleotide polymorphisms (SNP) and haplotypes of the *TNF*-block associated with HIV-SN. In addition to TNF-1031, we consider markers of the Caucasian 8.1 ancestral haplotype (AH; HLA A*0101: Cw*0701: B*0801: DRB1*0301: DQA1*0501: DQB1*0201), which has been associated with many immunopathological conditions including diabetes and coeliac disease [17]. The most studied candidate polymorphism lies at position -308 upstream of *TNF* (TNF-308; rs1800629). Despite several promising ex vivo studies, alleles of rs1800629 did not affect TNF production when analyzed using promotor constructs [18], so other SNP within the conserved haplotype may be important. As rs1800629*A occurs in several *TNF*-block haplotypes, we use an indel in intron 10 of the *BAT1/DDX39B* gene (rs9281523*C) as a specific marker of the *TNF*-block of the 8.1AH [14]. In Asians, the minor alleles of rs1800629 and rs9281523 occur as part of the diabetogenic 8.1 AH [19,20], consistent with a role for the region in immunopathology. The minor allele of rs9281523 associated with risk of HIV-SN in Caucasian patients who developed a toxic neuropathy following exposure to stavudine [6], but showed no effect in our cross-sectional studies of Indonesian or South African patients treated with stavudine [5,16]. Here, we assess two cohorts with no exposure to stavudine.

Parallel investigation in two cohorts of different ethnicity has potential to identify critical SNP within conserved haplotypes. Specifically, instances where the predominant haplotypes vary, but the same SNP associates with the phenotype provide circumstantial evidence that the SNP contributes directly to the phenotype.

## 2. Results

### 2.1. Measures of the Severity of HIV Disease Predict HIV-SN in Indonesians and Africans

The South African cohort (*n* = 75) included 29 patients with HIV-SN (38.7%). Participants were relatively young [39 (19–60)] and 60% were female (45/75). Patients with HIV-SN were a little heavier and taller than those without. Height, weight and lower nadir CD4 T-cell counts associated significantly (*p* < 0.05) with HIV-SN. A lower current CD4 T-cell count, a history of tuberculosis and >500 copies of HIV RNA/mL associated weakly (*p* < 0.20) with HIV-SN (Table 1). The regression model retained greater weight, tuberculosis and a low nadir CD4 T-cell count independently associated with HIV-SN (model *p* = 0.0007; pseudo *R*^2^ = 0.18; Appendix A).

The Indonesian cohort (*n* = 202) included 34 patients with HIV-SN (16.8%). Participants were relatively young [35 years (19–60 years)] and 29% were female (58/202). A lower current CD4 T-cell count and >500 copies of HIV RNA/mL were significantly associated with HIV-SN in bivariate analyses (*p* < 0.05; Table 1). A lower nadir CD4 T-cell count and a history of tuberculosis were weakly linked with HIV-SN (*p* < 0.20) and were also included multivariate analyses. The regression model identified a current viral load >500 copies of HIV RNA/mL and a lower current CD4 T-cell count as independently associated with HIV-SN (*p* = 0.0006; pseudo *R*^2^ = 0.08; Appendix A) [21].

We note that height was associated with HIV-SN in African and Indonesian patients treated with stavudine. Here, in Indonesians, height was excluded from logistic regression modeling as it did not meet inclusion criteria (*p* = 0.71; Table 1). However, in Africans, height met criteria for inclusion in logistic regression modelling (*p* = 0.03; Table 1) but was not retained in the optimal model. It is unlikely that associations with body weight in the optimal model in Africans arise through correlations with height (Pearson’s *r* = 0.225, *p* = 0.06).

### 2.2. Two Alleles Associated with HIV-SN in Indonesians but not Africans

Thirteen SNP previously linked with HIV-SN either independently or within haplotypes were selected from across the *TNF*-block for genotyping in these cohorts (Table 2) [6,15,16]. In the Indonesian cohort, the minor alleles of rs9281523 (*DDX39B*/*BAT1*(intron10)) and rs1800629 (TNF-308) were more common in patients with HIV-SN. These alleles were in tight LD but the minor allele of rs1800629 also occurred alone, so only rs9281523*C was included in logistic regressions as it reflects carriage of both alleles. No other SNP attained *p* < 0.20—the cut-off for inclusion in logistic regressions. The optimal model included >500 copies HIV RNA/mL, a lower current CD4 T-cell count and the minor allele of rs9281523 (*p* = 0.0011, pseudo *R*^2^ = 0.09; Table 3).

Minor alleles of rs2981523 and rs1800629 were present at higher frequencies but were not associated with HIV-SN in Africans (Table 2). However weak associations with reduced risk (*p* < 0.20) were evident with rs4947324*T (between *NFKBIL1* and *LTA*) and rs1041981*C (in *LTA*). These were included in logistic regression modeling. The resulting model (*p* = 0.003, pseudo *R*^2^ = 0.23; Table 4) incorporated weight, tuberculosis, lower nadir CD4 T-cell count and rs4947324*T.

### 2.3. One Haplotype Containing rs9281523*C and rs1800629*A Associated with HIV-SN in Indonesians but Not Africans

FastPHASE generated 10 haplotypes in Indonesians and 12 haplotypes in Africans present at >1.0%. These accounted for 98% and 96% of each population, respectively (Table 5 and Table 6). Eight haplotypes were shared between the two populations (S1–S8) and are numbered in order of the frequency of each haplotype in Africans. An additional four haplotypes were unique to Africans (A1–A4) and two were unique to Indonesians (I1, I2).

The haplotype S2 contained the minor alleles of rs9281523 and rs1800629, and associated with HIV-SN in bivariate analyses of the Indonesian cohort (Table 5) but did not remain in the final model (Table 3). This haplotype was also common in Africans but was not linked with HIV-SN. Three other haplotypes (S1, A3 and S7) carried by Africans met the criterion for inclusion in logistic regression models (*p* < 0.20; Table 6). However, A3 and S7 perfectly predicted the absence of HIV-SN and could not be included. The resulting model (*p* = 0.0003, pseudo *R*^2^ = 0.22; Table 4) included greater body weight, prior TB, lower nadir CD4 T-cell counts and S1. S1 contains no minor alleles.

### 2.4. One Haplogroup Contained the Two Haplotypes Associated with HIV-SN in Africans and Indonesians

A haplotype network was constructed for the 14 haplotypes occurring at >1% in Africans or Indonesians; S1–S8, A1–A4 and I1–I2. The haplotype network identified two haplogroups (A and B), where each contained two shared haplotypes and one unique to Africans (Figure 1).

Haplogroup A included S1, S2 and A3—carried by 91% of the Africans and 50% of Indonesians. S1 associated with HIV-SN in Africans and S2 with HIV-SN in Indonesians (Table 5 and Table 6). S1 and S2 differ only at rs9281523 (DDX39B/BAT1 (intron10)) and rs1800629 (TNF-308), marking the 8.1AH. A3 perfectly predicted the absence of HIV-SN in Africans.

Haplogroup B contained S4, S8 and A2, and was carried by 20% of Africans and 57% of Indonesians. S8 was also not observed in Africans with HIV-SN. These three haplotypes share seven minor alleles and differ at a further two loci: S8 and A2 include the minor allele of rs2071593 and A2 carries the minor allele of rs4947324, which are associated with reduced risk of HIV-SN in Africans (Table 2).

## 3. Discussion

Despite the discontinuation of stavudine, HIV-SN remains a common neurological complication of HIV disease, impacting 14% of Indonesians surveyed in the 2016 cross-sectional study [10] and 38% of Africans in this study. We describe associations between HIV-SN and demographic variables, clinical variables and SNP and haplotypes in the *TNF*-block which have previously been linked with HIV-SN in Caucasians, Asians and Africans.

Age and height were strong predictors of HIV-SN in Africans and Indonesians treated with stavudine [5,9]. However, without stavudine, markers of HIV disease severity were more clearly associated with HIV-SN. In African patients, the optimal markers were lower nadir CD4 T-cell counts, greater weight and a history of tuberculosis whilst low current CD4 T-cell counts and >500 HIV RNA copies/mL were the clearest associations with HIV-SN in Indonesians. As the African patients had received ART for only 6–8 months, current CD4 T-cell levels may not have stabilized and a clearer association with nadir CD4 T-cell counts is reasonable. In Africans, tuberculosis remained independently associated with HIV-SN in all regression models (Appendix A and Table 4), despite being present in 41% of Indonesians (*cf* 15% of Africans) and associated weakly with HIV-SN in both populations in bivariate analyses (Table 1). Pyridoxine is routinely provided to patients treated for tuberculosis in South Africa and Indonesia, but a previous South African study linked inadequate levels of pyridoxine with neuropathy despite administration of supplements [22]. Overall, biological markers of HIV disease were the clearest demographic and clinical associations of HIV-SN. This is consistent with findings prior to the advent of ART (and hence, stavudine) [23] and suggests the underlying mechanisms of HIV-SN may differ between patients treated with and without stavudine.

Differences in the pathogenic pathways are also evidenced by the different genetic associations identified in this study. We found no associations between the minor allele of rs1799964 (TNF-1031) and HIV-SN in Africans or Indonesians evident in our studies of Asians and Caucasians treated with stavudine [5,6,15]. In Indonesians treated without stavudine, rs9281523*C (*DDX39B*/*BAT1* (intron10)) is independently associated with HIV-SN after correcting for lower current CD4 T-cell counts and >500 copies of HIV RNA/mL (model *p* = 0.0011, Pseudo *R*^2^ = 0.09; Table 3). The S2 haplotype, which contains rs9281523*C, was associated with HIV-SN in Indonesians in bivariate analyses. rs9281523*C marks the 8.1AH which has been linked with accelerated loss of CD4 T-cells and impaired recovery following HIV infection [24,25] and with numerous immunopathological diseases [17,26]. So, a link with HIV-SN is plausible and the search for the SNP responsible has wide ramifications. A direct role for rs9281523 is supported by its inclusion in the optimal model for Indonesians, while the associated haplotype (S2) was excluded. However, S2 and S1 (the haplotype linked with risk of HIV-SN in Africans) differed only at rs9281523 and rs1800629 and occurred in the same haplogroup (Figure 1). This suggests the haplotypes may have descended from a common ancestor, but argues against a direct role for rs9281523 or rs1800629. It also fits with the observation that rs9281523 shows no associations with HIV-SN in Africans.

In Africans, the minor alleles of rs4947324*T and rs1041981*C associated weakly with reduced risk of HIV-SN in bivariate analyses (Table 2). Both SNP have been associated with reduced risk of HIV-SN in South Africans treated with stavudine where rs1041981*C and six additional SNP from the *TNF*-block were included in the logistic regression model with age and height [16]. Without stavudine, rs4947324*T alone remained in the optimal model (*p* = 0.0003, Table 4). Three haplotypes (S1, A3 and S7) were weakly associated with altered risk of HIV-SN (*p* < 0.20; Table 6). A3 and S7 were rare haplotypes with no minor alleles in common, but S7 contained rs4947324*T and rs1041981*C. Neither A3 nor S7 occurred in patients with HIV-SN so they could not be included in logistic regression modeling. S1, was independently associated with risk of HIV-SN in the final model along with greater weight, tuberculosis and nadir CD4 T-cells/µL (*p* = 0.0003, Pseudo *R*^2^ = 0.22; Table 4). S1 included no minor alleles and therefore contained the major (*C) risk allele of rs4947324.

The search for SNP responsible for associations with *TNF*-block haplotypes is complicated by the lack of relevant in vitro assays for chronic conditions. For example, haplotypes containing polymorphisms within the *TNF*-block were associated with altered TNF and lymphotoxin-alpha production in cultured mononuclear cells from Australian Caucasians but haplotypes containing rs9281523*C and rs4947324*T had no impact [27]. SNP within the *TNF*-block may also impact other MHC genes. The GTEX eQTL database reports altered expression of several immune-related genes within the MHC is associated with carriage of rs9281523*C or rs4947324*T including *C4A*, *HLA-C* and *MICB* (https://gtexportal.org/). However, understanding the biological consequences of these SNP and associated haplotypes is complicated by LD within the region [28].

We recognize the small size of our cohorts, but strict inclusion criteria avoided alternative causes of neuropathy. Further studies are required to validate our findings in larger cohorts when they become available, and to address the biological implications. Overall, we confirm that HIV-SN remains a clinically relevant problem in HIV+ patients treated without stavudine and is more common in those with African ancestry. We confirm that *TNF*-block genotypes associate with HIV-SN but show that different mechanisms are invoked with and without stavudine. Critical genotypes differ between Indonesians and Africans so the causative alleles remain unknown. However, the haplotypes associated with HIV-SN may descend from a common ancestor, and therefore, include allele/s not typed in our panel. Associations with *TNF*-block haplotypes per se confirm the inflammatory etiology of HIV-SN. Moreover, the clearer associations with HIV-SN seen in Africans (evidenced by stronger regression models) may plausibly align with the greater frequency of HIV-SN in Africans compared with Indonesians. This distinction warrants further investigation.

## 4. Materials and Methods

### 4.1. Participants and Phenotypes

HIV-positive adults (*n* = 185) who had used ART for at least 12 months with no exposure to stavudine were screened for neuropathy at POKDISUS HIV Care Clinic, Cipto Mangunkusumo Hospital, Jakarta, Indonesia in 2016 [21,29]. Patients with any history of other conditions potentially linked with neuropathy were excluded. DNA was also available from nine HIV+ patients with HIV-SN and eight patients without HIV-SN matched by age, gender and CD4 T-cell count. These 17 individuals were recruited in 2012 at the same clinic and met inclusion criteria for the present study. Patients with African ancestry attending the Lenasia South Community Health Hospital, Johannesburg, South Africa, were enrolled after 6–8 months on ART. These studies were approved by the Ethics Committee of the Faculty of Medicine, Universitas Indonesia (approval number: (579/UN2.F1/ETIK/2014), and the Human Research Ethics Committee (Medical) of the University of the Witwatersrand, Johannesburg (approval number: MR121018-R14/49). Written informed consent was obtained in all cases.

HIV-SN was assessed using the AIDS Clinical Trials Group Brief Peripheral Neuropathy Screening Tool, a validated metric for diagnosing HIV-SN [30]. HIV-SN was defined by bilateral presence of at least one symptom (pain/burning/aching, numbness, and paresthesia) and one clinical sign (absent reflexes and impaired vibration sense in the great toe). In South Africans, pinprick sensitivity was added to the BPNS as it provides an assessment specific to small fiber pathology and has high specificity for HIV-SN [31,32]. Clinical and demographic records were collected from medical files to determine risk factors that may be linked with HIV-SN. This included the patient’s history of tuberculosis as medications used in its treatment, notably isoniazid, competitively interferes with pyridoxine metabolism and may result in peripheral neuropathy [22,33].

### 4.2. Genotyping

As described previously [21], genomic DNA was extracted from EDTA-blood samples using Favorprep Blood Genomic DNA Extraction Mini Kits (Favorgen Biotech Corporation, Changzhi, Taiwan) adjusted to 50 ng/µL and diluted 1:1 in TaqMan^®^ OpenArray™ Genotyping Master Mix (Life Technologies, Grand Island, NY, USA). Samples were genotyped for 13 SNP (Table 2) spanning *MMCD1*, *DDX39B*, *ATP6V1G2*, *NFKBIL1*, *LTA* and *TNF* using the QuantStudio 12K Flex Real-Time PCR System on custom TaqMan^®^ OpenArrayTM Real-Time PCR Plates. All alleles were in Hardy-Weinberg Equilibrium (HWE).

### 4.3. Haplotype Analyses

Haplotypes and their estimated frequencies were determined using the default parameters of the fastPHASE algorithm [34], with haplotypes sampled from the observed genotypes an additional 5000 times per sample. Haplotypes with an estimated frequency less than 1% were excluded from analyses. Haplotypes shared between Africans and Indonesians are labelled S1–S8 in order of their population frequencies in Africans. Haplotypes unique to Africans are labelled A1–A4 and haplotypes unique to Indonesians are labelled I1 and I2 in order of their respective population frequencies. PopART version 1.7 (Population Analysis with Reticulate Trees, Otago, New Zealand; http://popart.otago.ac.nz) was used to construct haplogroups using Median-Joining methods [35].

### 4.4. Statistical Analyses

Bivariate associations between HIV-SN and demographic or clinical variables, SNP and haplotypes were assessed with *t*-tests, Mann-Whitney tests, Chi^2^ or Fisher’s exact tests using GraphPad Prism version 8.2.1 for Windows (Graphpad Software, La Jolla, CA, USA). No corrections were made for multiple comparisons. All variables which weakly (*p* = 0.05–0.20) or significantly (*p* < 0.05) associated with HIV-SN in bivariate analyses were included in logistic regression modelling. Optimal models identifying variables independently associated with HIV-SN were determined with a stepwise removal process using Stata/IC 16.0 for Windows (StataCorp LLC, College Station, TX, USA).

## Figures and Tables

**Figure 1 ijms-21-00380-f001:**
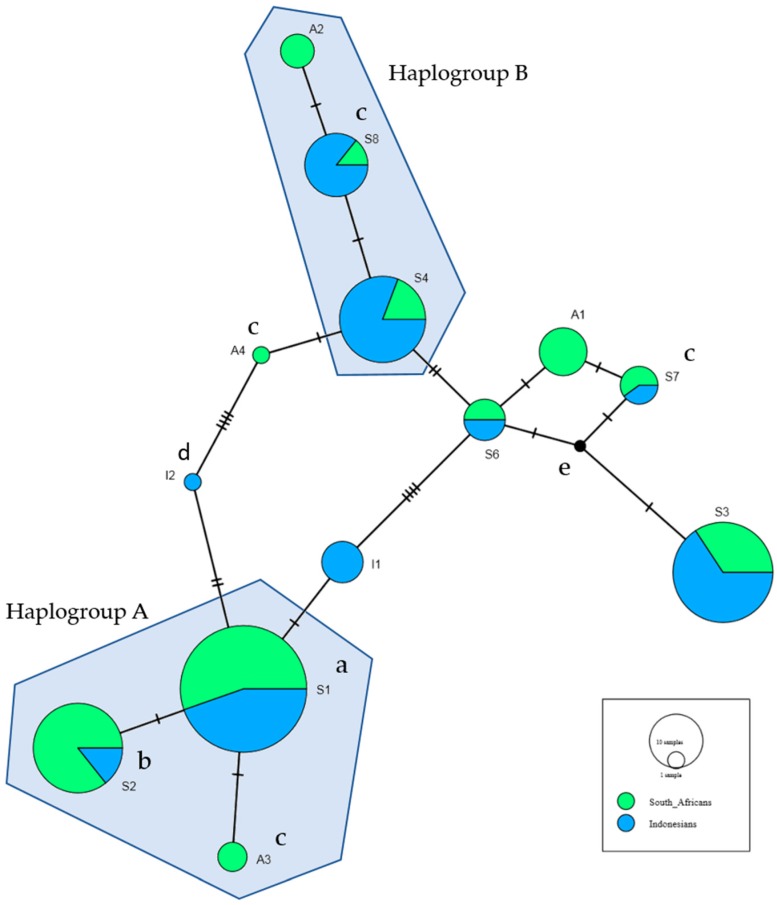
The haplotype network was constructed using all haplotypes which occurred in Africans and Indonesians at greater than 1%. This includes the eight shared haplotypes, four haplotypes unique to Africans and two haplotypes unique to Indonesians, as defined as in Table 5 and Table 6. ^a^ Associated with increased risk of HIV-SN in Africans in bivariate analyses (Table 6). ^b^ Associated with increased risk of HIV-SN in Indonesians in bivariate analyses (Table 5). ^c^ Haplotypes not found in Africans with HIV-SN. ^d^ Haplotypes not found in Indonesians with HIV-SN. ^e^ Median vector: hypothetical haplotype automatically generated for maximum parsimony.

**Table 1 ijms-21-00380-t001:** HIV-SN associates with CD4 T-cell counts and control of HIV replication.

Variable	Africans			Indonesians		
	HIV-SN			HIV-SN		
	+ve (*n* = 29)	−ve (*n* = 46)	*p* ^a,b^	+ve (*n* = 34)	−ve (*n* = 167)	*p* ^a,b^
Age (years)	40 (24–60)	37 (19–58)	0.11	36 (21–59)	35 (19–60)	0.68
Height (cm)	**168 (147–179)**	**163 (135–186)** ***n* = 45**	**0.03**	167 (151–185)	167 (142–180) *n* = 166	0.71
Weight (kg)	**66 (45–112)**	**55 (35–110)** ***n* = 44**	**0.03**	59 (39–88)	58.5 (37–104) *n* = 166	1.00
Current CD4 T-cells/µL	221 (22–685)	300 (8–832)	0.06	**326 (44–729)**	**458 (84–1166)**	**0.003**
Nadir CD4 T-cells/µL	**107 (4–575)**	**223 (8–771)**	**0.002**	54 (3–428)	121 (1–599) *n* = 165	0.06
HIV RNA >500 copies/mL	21/29 (72%)	25/46 (54%)	0.12	**6/29 (17%)**	**7/163 (4.1%)**	**0.005**
History of Tuberculosis	6/28 (29%)	3/45 (7%)	0.08 ^c^	18/35 (53%)	66/168 (39%)	0.09
Female Gender	15/29 (52%)	30/46 (65%)	0.25	9/35 (26%)	49/167 (29%)	0.98

^a^ Mann-Whitney test used to assess all continuous variables—Median (range); ^b^ χ^2^ test used to assess dichotomous variables—Proportion (%); ^c^ Fisher’s Exact test used where *n* < 5; Shading marks factors included in logistic regressions. Significant differences are shown in bold font.

**Table 2 ijms-21-00380-t002:** Alleles of two SNP associate with HIV-SN in Indonesians but not Africans.

SNP rsID	Africans (*n* = 75)					Indonesians (*n* = 202)				
	Minor Allele	MAF ^a^	HIV-SN		*p* ^e^	Minor Allele	MAF ^a^	HIV-SN		*p* ^d^
			+ve ^b^	−ve				+ve ^b^	−ve ^c^	
rs2075582 (*MCCD1*)	C	0.13	7/28 ^d^ (25%)	12/46 (27%)	0.87	C	0.34	19/34 ^e^ (56%)	94/166 (57%)	0.97
rs9281523 (*DDX39B*)	-	0.20	9/29 (31%)	17/46 (37%)	0.60	-	0.03	**5/29 (17%)**	**8/166 (5%)**	**0.03**
rs11796 (*DDX39B*)	A	0.41	16/29 (55%)	31/46 (67%)	0.27	T	0.35	21/35 (60%)	95/167 (57%)	0.73
rs2523506 (*DDX39B*)	T	0.12	6/28 (21%)	9/46 (20%)	0.85	T	0.34	13/34 (38%)	70/165 (42%)	0.65
rs2523504 (*intergenic*)	T	0.17	8/29 (28%)	15/46 (33%)	0.60	T	0.33	19/35 (54%)	92/165 (56%)	0.87
rs2071594 (*intergenic*)	G	0.41	16/29 (55%)	31/46 (67%)	0.29	C	0.37	21/35 (60%)	98/165 (59%)	0.95
rs2071593 (*intergenic*)	A	0.07	2/29 ^f^ (7%)	8/46 (17%)	0.30	A	0.12	7/34 (21%)	38/167 (23%)	1.00
rs2071592 (*NFKBIL1*)	T	0.42	16/29 (55%)	30/46 (67%)	0.32	A	0.30	19/34 (56%)	83/166 (50%)	0.53
rs4947324 (*intergenic*)	T	0.16	6/29 (21%)	16/46 (35%)	0.19	T	0.03	3/35 (9%)	8/166 (5%)	0.41
rs909253 (*LTA*)	A	0.41	16/29 (55%)	31/46 (67%)	0.29	G	0.36	21/35 (60%)	99/167 (59%)	0.94
rs1041981 (*LTA*)	C	0.39	15/29 (52%)	30/46 (67%)	0.20	A	0.36	21/35 (60%)	99/167 (59%)	0.94
rs1799964 (*intergenic*)	C	0.15	6/29 (21%)	12/46 (27%)	0.56	C	0.27	16/35 (46%)	76/165 (46%)	0.97
rs1800629 (*intergenic*)	A	0.23	10/29 (34%)	19/46 (41%)	0.56	A	0.04	**6/35 (17%)**	**11/167 (7%)**	**0.04**

^a^*MAF*: minor allele frequency; ^b^ Individuals with HIV-SN who carry one or two copies of the minor allele; ^c^ Individuals without HIV-SN who carry one or two copies of the minor allele; ^d^ χ^2^ or Fisher’s Exact test if *n* < 5; ^e^ Up to six samples failed to genotype for each SNP; ^f^ Two SNP in Indonesians which associated with HIV-SN (*p* < 0.05) are in bold; Two SNP in Indonesians and two SNP in Africans which met the criteria for inclusion in logistic regression modelling (*p* < 0.20) are shaded.

**Table 3 ijms-21-00380-t003:** Logistic regression modelling identified rs9281523*C in Indonesians as an independent marker of susceptibility to HIV-SN.

Variable	Odds Ratio	*p* Value	95% Confidence Interval
**SNP Model:***n* = 193 ^a^, *p* = 0.0011, Pseudo *R*^2^ = 0.09			
Current CD4 T-cells/µL	1.00	0.02	1.00–1.00
>500 copies HIV RNA/mL	1.86	0.12	0.85–4.07
rs9281523*C	2.49	0.15	0.71–8.65
**Haplotype Model:** No haplotypes independently associated with HIV-SN after correction for current CD4 T-cells/µL and >500 copies of HIV RNA/mL			

^a^ Excluding samples with missing demographic, clinical and/or genotype data.

**Table 4 ijms-21-00380-t004:** Logistic regression modelling identified rs4947324*T in Africans as an independent marker of susceptibility to HIV-SN.

Variable	Odds Ratio	*p* Value	95% Confidence Interval
**SNP Model:***n* = 71 ^a^, *p* = 0.0003, Pseudo *R*^2^ = 0.23			
Weight (kg)	1.04	0.04	1.00–1.08
History of Tuberculosis	5.66	0.04	1.09–29.36
Nadir CD4 T-cells/µL	0.99	0.02	0.99–1.00
rs4947324*T	0.25	0.05	0.06–1.01
**Haplotype Model:***n* = 71 ^a^, *p* = 0.0003, Pseudo *R*^2^ = 0.22			
Weight (kg)	1.04	0.02	1.01–1.08
History of Tuberculosis	5.22	0.04	1.09–24.86
Nadir CD4 T-cells/µL	0.99	0.02	0.99–1.00
S1 (Shared Haplotype 1)	3.21	0.07	0.93–11.12

^a^ Excluding samples with missing demographic, clinical, genotype data and/or haplotypes perfectly aligned with the absence of HIV-SN.

**Table 5 ijms-21-00380-t005:** One haplotype associated with HIV-SN in Indonesians.

Haplotype ^a^	rs2075582	rs9281523	rs11796	rs2523506	rs2523504	rs2071594	rs2071593	rs2071592	rs4947324	rs909253	rs1041981	rs1799964	rs1800629	HIV-SN ^b^				*p* ^c^
														+ve (*n* = 35)		−ve (*n* = 167)		
S1	T	-	T	G	C	C	G	A	C	G	A	T	G	15	43%	72	43%	0.69
S3	T	-	A	T	C	G	G	T	C	A	C	C	G	14	40%	70	42%	0.85
S4	C	-	A	G	T	G	G	T	C	A	C	T	G	12	34%	60	36%	0.87
S8	C	-	A	G	T	G	A	T	C	A	C	T	G	7	20%	36	22%	0.97
I1	T	-	T	G	C	C	G	T	C	G	A	T	G	3	9%	21	13%	0.77
S6	T	-	A	G	C	G	G	T	C	A	C	T	G	2	6%	10	6%	0.99
**S2 ^d^**	T	C	T	G	C	C	G	A	C	G	A	T	A	**5**	**14%**	**8**	**5%**	**0.02**
S7	T	-	A	G	C	G	G	T	T	A	C	C	G	2	6%	8	5%	0.66
I2	C	-	A	G	C	C	G	A	C	G	A	T	G	0	0%	5	3%	0.99
S5	T	-	T	G	C	C	G	A	C	G	A	T	A	1	3%	3	2%	0.51

^a^ Haplotypes shared between Africans and Indonesians are labelled S1–S8 in order of their population frequencies in Africans. Haplotypes unique to Indonesians are labelled I1 and I2 in order of population frequencies. Haplotypes carried at <1% are excluded. Minor alleles for each population are shaded grey; ^b^ Number of individuals who carry one or two copies of each haplotype; ^c^ χ^2^ (or Fisher’s Exact test if *n* < 5); ^d^ Haplotypes meeting logistic regression criteria (*p* < 0.20) are in bold.

**Table 6 ijms-21-00380-t006:** Three haplotypes were weakly associated with HIV-SN in Africans.

Haplotype ^a^	rs2075582	rs9281523	rs11796	rs2523506	rs2523504	rs2071594	rs2071593	rs2071592	rs4947324	rs909253	rs1041981	rs1799964	rs1800629	HIV-SN ^b^				*p* ^c^
														+ve (*n* = 29)		−ve (*n* = 46)		
S1 ^d^	T	-	T	G	C	C	G	A	C	G	A	T	G	**19**	**66%**	**23**	**50%**	**0.19**
S2	T	C	T	G	C	C	G	A	C	G	A	T	A	9	31%	16	35%	0.34
S3	T	-	A	T	C	G	G	T	C	A	C	C	G	5	17%	8	17%	0.99
A1	T	-	A	G	C	G	G	T	T	A	C	T	G	3	10%	8	17%	0.51
S4	C	-	A	G	T	G	G	T	C	A	C	T	G	3	10%	5	11%	0.99
S5	T	-	T	G	C	C	G	A	C	G	A	T	A	1	3%	4	9%	0.64
A2	C	-	A	G	T	G	A	T	T	A	C	T	G	1	3%	4	9%	0.64
S6	T	-	A	G	C	G	G	T	C	A	C	T	G	3	10%	2	4%	0.37
A3	T	-	T	G	T	C	G	A	C	G	A	T	G	**0**	**0%**	**4**	**9%**	**0.15**
S7	T	-	A	G	C	G	G	T	T	A	C	C	G	**0**	**0%**	**4**	**9%**	**0.15**
S8	C	-	A	G	T	G	A	T	C	A	C	T	G	0	0%	2	4%	0.52
A4	C	-	A	G	T	G	G	T	C	A	A	T	G	0	0%	2	4%	0.52

^a^ Haplotypes shared between Africans and Indonesians are labelled S1–S8 in order of their population frequencies in Africans. Haplotypes unique to Africans are labelled A1–A4 in order of population frequencies. Haplotypes carried at <1% are excluded. Minor alleles for each population are shaded grey; ^b^ Number of individuals who carry one or two copies of each haplotype; ^c^ χ^2^ (or Fisher’s Exact test if *n* < 5); ^d^ Haplotypes meeting logistic regression criteria (*p* < 0.20) are in bold.

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
