# Peer review of "TNF-Block Genotypes Influence Susceptibility to HIV-Associated Sensory Neuropathy in Indonesians and South Africans"

_ijms, 2020, doi:10.3390/ijms21020380_

Round 1
Reviewer 1 Report
1.Genetic studies in Humans are inherently problematic with the most common problem being the difficulty of correcting for multiple comparisons when many loci or SNPs are being tested or when there are multiple alleles at a selected locus (such as HLA). Bonforoni corrections are too severe to be practical. A more reasonable approach is observe involvement of a given allele, haplotype, or SNP in a primary population and then test that hypothesis that the correlation exists in a non overlapping matched population.
This manuscript might be a failed attempt of the later method. Although each population analysis in this manuscript is internally reasonable and interesting, comparisons of the two ethnic groups are problematic given 1) lack of control gene locus (non TNF genes), 2) controlling for different characteristics in the two populations (HIV RNA>500 and TB history in one and not the other) and 3) the lack of ex-vivo data supporting a mechanism (such as higher TNFa or receptor expression with identified SNP). Together, these deficiencies undermine the comparisons between the two ethic populations, especially since the methods say that correction for multiple comparisons is unnecessary because the data are “exploratory”. If it is exploratory, then there should be mention of the need for second African and Indonesian populations to confirm genetic associations and comparisons are premature.
The presence of stavudine in the title and abstract when the data does not involve stavudine does not help this manuscript. HIV associated SN existed before stavudine and exists after stavudine. While it is understandable that the authors discuss stavudine associations in the introduction because of historical findings, objectively there does not seem to be a good reason to include it in the title or abstract of this manuscript. It “dates” the manuscript.
The associations between SN and severity of HIV disease are of interest. Rather than controlling for HIV RNA levels, the authors might consider dividing the populations into HIV RNA detectable and non detectable to see if genetic associations are stronger or weaker when HIV RNA is suppressed. If the SN is actively triggered by HIV replication, one would predict that the association would be stronger in HIV RNA detectable subjects. Also, separate analysis of the HIV RNA suppressed individuals might reveal age effects.
Overall, the manuscript would be more acceptable if the authors:
Did not attempt direct comparison between the 2 ethnic groups because it is an exploratory study and Stated that a comparison was not appropriate without a second ethnic group for each to confirm different associations Reduced the mention of stavudine in title and abstract.Author Response
See attached file

Reviewer 2 Report
In the manuscript entitled “Ethnicity and ART without stavudine influence associations between HIV-associated sensory neuropathy and TNF-block genotypes”, authors investigated associations between HIV-SN and demographic characters as well as the genotypes in TNF-block in the patient cohorts without stavudine treatment. Their data suggest that in the South African cohort, height, weight and Nadir CD4 T-cells significantly associated with HIV-SN; while in the Indonesian cohort, a lower current CD4 T-cell count and >500 copies of HIV RNA/ml were significantly associated with HIV-SN. The authors also examined the association between genotypes in the TNF-block and HIV-SN and found two positive alleles in the Indonesian cohort but not the Africans.
In general, this study shows some interests to nail down possible contributing host factors of HIV-SN, while the data shown here are not very convincing.
Major concerns:
Table 2, the p<0.05 is not a convincing standard for a strong association between a genotype and phenotype, especially considering the relatively small sample size. Also, in the South African cohort, the two alleles, rs9281523 and rs1800629, showed opposite pattern as compared to that in the Indonesian cohort, which make it even suspicious whether there is actual association between these two alleles and the HIV-SN. Authors didn’t discuss what’s the possible mechanism of these associations and HIV-SN, especially in the context of the difference between two cohorts. For example, how could height and weight contribute to HIV-SN in African cohort, while these factors are less influential in the Indonesian cohort. The same question is true for the associated alleles, what’s the possible gene regulating pathway these alleles involved could contribute to the progress of HIV-SN, is there any experiments could test the hypothesis?
Round 2
Reviewer 1 Report
The revision is a great improvement. The change in title and the abstract reveal the true focus of the study and will be more likely to attract a wider scope of readers. The clarifications throughout the manuscript and the more cautious statements allow the readers better make their own judgments concerning the data.
Reviewer 2 Report
My concerns have not been properly addressed.